# Association of Social Determinants with Patient-Reported Outcomes in Patients with Cancer

**DOI:** 10.3390/cancers16051015

**Published:** 2024-02-29

**Authors:** Hollis Hutchings, Parnia Behinaein, Nosayaba Enofe, Kellie Brue, Samantha Tam, Steven Chang, Benjamin Movsas, Laila Poisson, Anqi Wang, Ikenna Okereke

**Affiliations:** 1Department of Surgery, Henry Ford Health, Detroit, MI 48202, USA; hollis.hutchings@gmail.com; 2School of Medicine, Wayne State University, Detroit, MI 48202, USA; parnia.behinaein@wayne.edu (P.B.); kellie.brue@luhs.org (K.B.); 3Division of Surgical Oncology, Department of Thoracic Surgery, Fox Chase Cancer Center, Temple University Hospital, Philadelphia, PA 19140, USA; enofe.nosa@gmail.com; 4Department of Otolaryngology, Henry Ford Health, Detroit, MI 48202, USA; stam2@hfhs.org (S.T.); schang1@hfhs.org (S.C.); 5Department of Radiation Oncology, Henry Ford Cancer Institute, Detroit, MI 48202, USA; bmovsas1@hfhs.org; 6Department of Public Health Sciences, Henry Ford Health, Detroit, MI 48202, USA; lpoisso1@hfhs.org (L.P.); awang7@hfhs.org (A.W.)

**Keywords:** cancer, distress, social determinants, disparities, survival

## Abstract

**Simple Summary:**

Patient-reported outcome (PRO) scores can help to reduce barriers between patient expectations and caregiver perspectives. The relationship between PRO scores and social determinants has not been studied widely. At a single large urban institution, all patients with a new cancer diagnosis who completed a survey over a two-year period were included. The PRO survey recorded perceived outcomes for depression, fatigue, pain interference and physical function. Over the two-year period, 1090 patients were included. Married patients had overall better PRO scores for each domain. Patients who had the ability to use the online portal also had overall better PRO scores. Male patients and minority patients had worse pain scores. PRO scores varied by cancer site and in association with specific social determinants. Future interventions can address patient support by utilizing these findings.

**Abstract:**

Patient-reported outcome (PRO) scores have been utilized more frequently, but the relationship of PRO scores to determinants of health and social inequities has not been widely studied. Our goal was to determine the association of PRO scores with social determinants. All patients with a new cancer diagnosis who completed a PRO survey from 2020 to 2022 were included. The PRO survey recorded scores for depression, fatigue, pain interference and physical function. Higher depression, fatigue and pain scores indicated more distress. Higher physical condition scores indicated improved functionality. A total of 1090 patients were included. Married patients had significantly better individual PRO scores for each domain. Patients who were able to use the online portal to complete their survey also had better individual scores. Male patients and non-White patients had worse pain scores than female and White patients, respectively. Patients with prostate cancer had the best scores while patients with head and neck and lung cancer had the worst scores. PRO scores varied by cancer disease site and stage. Social support may act in combination with specific patient/tumor factors to influence PRO scores. These findings present opportunities to address patient support at institutional levels.

## 1. Introduction

Following a diagnosis of cancer, patients can experience varying levels of distress and anxiety [1,2,3]. In addition, patients may experience pain, changes in energy level and other disturbances to their well-being. Contemporary studies have indicated that the constellation and trajectory of these symptoms may be a predictor of mortality [4,5]. In the past, caregivers have elicited a patient history and performed unilateral examinations to determine prognosis and create a treatment plan. More recently, the medical community has relied on patient-reported outcome (PRO) scores to place more emphasis on patient perspectives. This shift to PRO score utilization has occurred to determine more accurately a patient’s perception of their quality of life and specific concerns after a cancer diagnosis. Utilizing PRO scores allows the medical community to manage the symptoms and distress that occur as a result of the diagnosis. In a team approach to health care, discrepancies between physician and patient perspectives can often go unrecognized. These barriers in communication can be exacerbated by numerous sociodemographic factors [6,7]. The utilization of PRO surveys may help to mitigate these communication lapses. The previous literature has shown that targeting interventions based on PRO scores can result in improved patient quality of life, a better ability to tolerate definitive treatment and decreased complications from treatment [8,9]. When combined with traditional history and physical examination, PRO scores may help to target resources and lessen suffering in patients with cancer. PRO scores can be used to advance equity in health care by offering patients the ability to express their concerns and experience with a measurable, validated tool, decreasing the probability of bias affecting results or physicians’ comprehension [10]. The use of PRO scores is relatively limited, however. Only a small percentage of new cancer patients ever have PRO scores measured or interventions based on PRO scores [11,12,13].

Social determinants of health refer to the social, economic and environmental conditions in which people are born, grow, live, work and age. These factors can have a profound impact on physical, mental and social well-being [14]. These conditions include access to resources and services, the physical and social environment, quality of housing and availability of health care and education [15]. Approximately twenty percent of health outcomes can be attributed to clinical care, whereas socioeconomic factors, health behaviors and the physical environment have a more substantial effect [16]. As an example, African Americans tend to have worse survival for most cancers in the United States [17]. There has been only sparse literature concerning the relationship of social determinants of health with patient-reported views [18]. Disparities in treatment may lead to varying levels of distress experienced by patients. African Americans and economically disadvantaged patients tend to be diagnosed at advanced stages compared to other races and have greater obstacles to obtaining appropriate treatment [19,20,21]. As such, our goal was to examine the association of social determinants and other patient factors with PRO scores.

## 2. Materials and Methods

### 2.1. Patients

This study was performed at Henry Ford Health (Detroit, MI, United States), a tertiary cancer center located in the center of a large urban community. The medical center is situated in the northern United States and has been present for several decades in its current location. The cancer center serves a diverse population and treats thousands of patients with a new diagnosis of cancer annually. Institutional review board approval was obtained prior to conducting the study. All patients with a diagnosis of cancer were eligible to complete a PRO survey at one large urban/suburban health system. While all patients were eligible, no patient was required to complete the survey. The PRO surveys were automatically assigned to the electronic health record of patients who were at least 18 with an oncologic International Classification of Diseases, 10th Revision, clinical modification code on their problem list and seeing a surgeon, radiation oncologist, medical oncologist or oncology support provider. PRO surveys were available and offered prior to their visit by using the electronic health record or in person. When in person, tablets were provided to each patient to complete the survey. The PRO survey was selected by an internal patient-reported outcomes committee. The committee comprised members from the following specialties: medical oncology, radiation oncology, surgery, palliative care medicine, public health sciences and social work. The survey used was obtained from the National Institute of Health’s Patient-Reported Outcomes Measurement Information System (PROMIS) [22]. The PROMIS computer adaptive test (CAT) was chosen because of its user friendliness and ability to measure specific domains with relatively few questions. All patients who chose to complete the form had their information stored in a secure database.

### 2.2. PRO Survey

The PRO survey contained four separate domains that were scored: physical function, fatigue, pain interference and depression. Each domain was scored between 0 and 100. A higher category score in fatigue, pain interference and depression indicated a higher level of distress. Conversely, a higher physical function score indicated less distress. Selected PRO measures were domain-specific, rather than disease-specific, to allow for the application of the survey across all primary cancer sites. These measures were developed and validated to measure function and symptoms in both the general population and those with chronic conditions like cancer. Patients had the option to complete the survey through their online medical record portal if they desired and had the means to complete the survey electronically. If unable to complete via their online medical portal, patients were given the opportunity to complete the survey at their clinic visit by using designated iPads.

All patients with a new cancer diagnosis between 2020 and 2022 were included in this study. A patient was included if the PRO survey was completed within 6 months of the initial date of diagnosis. Patient demographics, including race, gender, marital status, insurance type, Charlson Co-Morbidity Index (CCI) and race, were included. For employment level and income, zip-code-level statistics were used for each patient and were obtained from the United States 2020 census tabulation [23]. Tumor characteristics, such as the disease’s primary site and stage, were recorded. The method of completing the survey (online vs. in person) was also recorded.

### 2.3. Statistical Analysis

The continuous measures were summarized as means with standard deviation. The categorical data were summarized by percentage. Differences in the means between the groups were compared by a *t*-test or analysis of variance (ANOVA). Significant ANOVA results were followed by post hoc *t*-tests between the factor levels. Associations between the continuous variables were measured by Spearman’s correlation coefficient (rho). The 95% confidence interval is presented for rho, with a *p*-value for the significant differences from rho = 0 computed by an asymptotic t approximation. Variable selection for multivariable models of the PROMs used the importance scores from random forest regression. For PROMs that had multiple correlated census-level variables of high importance, the variable(s) with importance across the PROM domains was retained for easier comparison. Multivariable modeling was conducted by linear regression. All significance testing used an alpha = 0.05 threshold for two-sided hypothesis tests. All the analyses were completed on patient-level data. To understand the patient population, descriptive statistics were reported.

## 3. Results

### 3.1. Demographics

A total of 7285 recorded surveys were completed by 4016 unique patients. Of this group, 1090 unique patients completed the survey within the first 6 months after their initial diagnosis and were included in this study (Table 1). The mean age was 60.2 ± 12.75 years. Female patients composed 60.3 percent of the cohort. There was a diverse patient population, with non-White patients representing 29.6% of the cohort. The majority of patients did not have private insurance, and nearly 80% of patients utilized the online portal to complete the survey.

### 3.2. Univariate Analyses of Social Determinants of Health versus PRO Scores

Table 2 shows univariate analyses of social determinants of health versus PRO scores. There was a significant association between decreasing age and improved physical function (approx. t, *p* < 0.01). Men had significantly higher pain interference compared to women (*t*-test, *p* = 0.04). Increased comorbidity burden was associated with worse fatigue, pain interference and physical function (approx. t, *p* < 0.01). Pain interference and physical function were improved for patients who were White compared to non-White patients, married patients compared to patients in other relationship states (ANOVA, *p* < 0.01), patients with private insurance compared to Medicaid and Medicare (ANOVA, *p* < 0.01), in neighborhoods with higher incomes (approx. t, *p* < 0.01) and in neighborhoods with decreased unemployment rates (approx. t, *p* < 0.01). Interestingly, patients who used the online portal to complete their survey had significantly improved individual PRO scores in each domain compared to those who completed their survey at their clinic visit (*t*-test, *p* < 0.01). Married patients and patients who utilized the online portal were among the patients who had significantly improved PRO scores.

### 3.3. Univariate Analyses of Tumor Characteristics versus PRO Scores

Table 3 shows univariate analyses of the tumor characteristics versus PRO scores. The mean PRO scores were the worst for patients with lung, gastrointestinal and head and neck disease in each domain (ANOVA, *p* < 0.01). The mean PRO scores were the best for patients with prostate cancer. The average PRO scores were worse for Stage IV disease in each domain (ANOVA, *p* < 0.01).

### 3.4. Multivariate Analyses

Table 4 shows the mean change in the PRO score for each domain, adjusting for unemployment rate, disease stage and other important predictors. Patients with an advanced disease had significantly worse depression and fatigue scores. Pain interference scores were significantly worse for patients with head and neck and lung primary tumors compared to patients with breast tumors. Patients with prostate cancer, conversely, had significantly better pain interference scores than patients with breast cancer. Patients who completed their surveys online also had significantly better pain scores. Physical function scores were worse for patients with advanced disease and patients with Medicare or Medicaid insurance compared to those with private insurance. Finally, physical function scores decreased with increasing comorbidity burden.

## 4. Discussion

Patient-reported outcomes have become a more valuable tool in treating patients diagnosed with cancer. Patients diagnosed with cancer demonstrate varying levels of stress and anxiety associated with their disease. By using a patient-reported outcome measurement, providers are better able to address the specific areas of distress and customize care to the patient. Our goal was to examine the associations of social determinants of health with patient-reported outcomes. Based on our results, it appears that there are significant interactions between social determinants of health and fatigue, pain interference, physical function and depression PRO scores among patients being treated for a new cancer diagnosis. Understanding the relationship between these social determinants of health and PRO scores will help caregivers guide management more effectively.

To our knowledge, our institution was one of the first cancer programs to use real-time PRO scores as a “vital sign” for cancer care across the entire system [24]. In our practice, PRO scores that meet the threshold for a “severe” issue are transmitted immediately to a team that can refer patients for needed interventions during the clinic visit. This immediate referral can improve quality of life, prevent unnecessary visits to the emergency room and reduce anxiety during the treatment phase. In review of the literature, recent collaborations have shown that monitoring PRO scores in real time can reduce unscheduled health services via supportive care interventions [25].

One interesting finding in our study was that patients who completed their surveys online by using the medical record portal had better scores. The ability to complete a survey online may possibly be a surrogate for improved social support and economic means. The previous literature has shown that patients with higher salaries, patients who live in a technology-dense neighborhood with high-speed internet and nonminority patients tended to use online medical record portals more frequently than other patients [26,27,28,29]. Despite increased availability over time, disparities in internet and computer access persist. In a recent population-based study, 23 percent of youth in low-income neighborhoods and 18 percent of African American youth overall had no internet access [30]. Additionally, African Americans, Native Americans and Alaska Natives are each less likely to have digital access compared to White Americans. This lack of access is also seen in rural households when compared to those living in urban areas [31].

In our institution, the PRO survey was offered to all patients prior to their visit by using the online portal and, if not completed, in person on the day of the visit by using designated iPads. There may be value in using the ability to complete the survey electronically prior to the appointment as a factor that guides the need for social work or case management referral. There are no studies yet that have studied the outcomes of patients with cancer based on the propensity to complete a PRO survey electronically. In patients undergoing kidney transplantation, however, there was no difference in outcomes based on the use of the online medical record portal [32].

In our study, African American and other non-White patients tended to have more pain interference and worse physical function decline than White patients. Unfortunately, there are still pervasive misconceptions about the thresholds of pain versus race. As recently as 2016, a study involving medical students and residents revealed that approximately 50 percent of participants had at least one false belief about White patients compared to African American patients [33]. Those beliefs included ideas such as “Blacks’ skin is thicker than Whites” and “Blacks’ nerve endings are less sensitive than Whites.” Other studies have shown that African American cancer survivors tend to have more daily pain than White survivors [34,35,36,37]. Our findings and previous studies seem to indicate that the medical community may not be addressing pain in non-White patients with cancer as well as White patients. In the future, increased awareness of these disparities may help reduce pain in cancer patients.

Neighborhoods with lower income and higher unemployment rates experienced worse PRO scores across multiple domains. Unfortunately, these factors are often associated with obstacles to care. Such obstacles include a lack of transportation or access to medical centers, increased medical mistrust and financial difficulties [14,38]. Language barriers may also be a factor, as some neighborhoods with lower socioeconomic statuses have a higher percentage of patients who do not speak English proficiently [39]. Medical centers can mitigate some of these obstacles to care by providing resources if and when available. As an example, ensuring that there are adequate translation services available may reduce anxiety or allow patients to relay any issues with pain more accurately. Another example would be providing transportation vouchers or ride-sharing services to patients based on PRO scores. These services are available at our institution, and we encourage other institutions to create similar resources for patients if possible.

The primary tumor site was independently associated with PRO scores. Patients with head and neck, lung and gastrointestinal tumors had the worst PRO scores in our study, and this persisted for pain interference in multivariable analyses. This trend is likely related to the increased dysfunction caused by these specific tumors. We also observed that the PRO scores worsened as the tumor stage advanced. Increased tumor burden and sequelae of disease would be expected to cause increased dysfunction [40,41]. Given the findings of our survey, targeting patients with an advanced disease at presentation may be helpful in addressing distress early to ease symptoms and improve quality of life. There should be consideration toward reflexive social work referrals for patients with advanced tumors. Also, future studies should track PRO scores longitudinally as the stage advances to determine optimal times to intervene.

PRO surveys measure condition-specific and generic symptoms [42]. There are currently hundreds of PRO surveys available. In selecting the appropriate survey, it is important to ensure that it measures the desired outcomes. In our survey, we utilized a validated tool developed by the NIH PROMIS group [43,44,45]. By using a publicly available and validated tool, we can ensure our patient scores are reliable measures of symptoms and distress.

PRO scores have been used as a research tool in cancer for many years and have been shown to be a strong predictor of survival [46]. More recently, a randomized trial in patients with advanced cancer showed that using real-time PRO scores actually improved overall survival [47]. Based on these studies, implementing PRO scores across an entire cancer enterprise should be beneficial to patient care. Our current study focused on the initial phases of diagnosis and treatment. Future studies will aim to determine the impact that PRO score monitoring has on outcomes.

The coronavirus disease 2019 (COVID-19) pandemic altered healthcare delivery and cancer care dramatically during part of this study. During the first few months of 2020, our institution had several weeks in which we temporarily stopped elective surgical operations on some patients. The number of patients who elected to come to our institution decreased as well. But, this study was conducted over a 3-year period, and many of the disruptive aspects of the pandemic had subsided for most of the study.

This study has limitations. Firstly, our survey was not obligatory. As such, there may be a selection bias in the patients who decided to complete the survey. However, during the study period, we estimated a response rate of approximately 33 percent. That response rate is generally higher than most other survey-based studies in the literature. Also, our cohort included an uneven representation of patients based on primary disease sites. Future studies should focus on more uniform participation among primary disease sites. Another limitation is that our cohort comes from a single institution. A larger multi-institutional study would demonstrate whether our findings can be generalized. Finally, we chose to include all patients who completed the PRO survey within 6 months of their initial diagnosis. We chose this time frame because, as a medical center in a large urban area, we frequently encounter patients who have extended delays between the date of diagnosis and the date of first treatment. Other similar institutions in urban areas have also seen intervals as long as 121 days [48]. Our goal was to include this set of patients without creating a time frame that was excessively long. Although we did not have longitudinal PRO score data on most patients, it did not appear that the treatment or interventions affected these initial PRO survey scores.

## 5. Conclusions

Increasing focus on the social determinants of health in clinical care and population health models has demonstrated that these elements have a profound impact on the quality of healthcare and patient outcomes. Certain groups of patients may have more distress after a cancer diagnosis. Utilizing a PRO survey may help to identify specific ways to provide support and lessen that distress. By measuring the impact of health care on patient satisfaction and quality of life, patient-centered outcomes can be used to assess the effectiveness of the care provided and the overall value of medical services. Future studies should be performed to measure multiple PRO scores for each patient. A longitudinal approach may help to determine the effectiveness of specific support resources offered to patients after a cancer diagnosis.

## Figures and Tables

**Table 1 cancers-16-01015-t001:** Sociodemographic and health characteristics of patients in the study. Henry Ford Health, Detroit, Michigan, January 2020 to December 2022.

Age, Years (Mean ± SD)	60 ± 12.75
**Female**	60.3% (658/1090)
**Race**	
White	66.1% (721/1090)
African American	25.7% (280/1090)
Other	3.9% (43/1090)
Unknown	4.2% (46/1090)
**Marital status**	
Married	53.8% (586/1090)
Single	33.9% (369/1090)
Other	5.2% (57/1090)
Unknown	7.2% (78/1090)
**Insurance status**	
Medicare	41.4% (451/1090)
Private	35.4% (386/1090)
Medicaid	9.3% (102/1090)
Unknown	13.9% (151/1090)
**Online portal use**	79.3% (864/1090)
**Primary disease site**	
Breast	30.5% (332/1090)
Head and neck	16.6% (181/1090)
Lung	14.8% (162/1090)
Gastrointestinal	14.6% (159/1090)
Prostate	2.2% (24/1090)
Other	21.3% (232/1090)
**Stage**	
0/in situ	3.9% (43/1090)
I	24.5% (267/1090)
II	12.7% (138/1090)
III	14.4% (157/1090)
IV	21.4% (233/1090)
Unknown	23.1% (252/1090)

**Table 2 cancers-16-01015-t002:** Univariable analyses of sociodemographic characteristics in relation to PRO scores. Henry Ford Health, Detroit, Michigan, January 2020 to December 2022.

Variable	Depression	Fatigue	Pain Interference	Physical Function
**Age ***	−0.02 (−0.08, 0.04)	−0.01 (−0.08, 0.06)	−0.03 (−0.10, 0.04)	−0.15 (−0.21, −0.08)
	*p* = 0.48	*p* = 0.85	*p* = 0.46	***p* < 0.01**
**Gender ****				
Female	51.1 (9.3)	53.5 (10.4)	54.5 (9.9)	41.8 (10.4)
Male	50.8 (9.3)	53.1 (11.2)	56.0 (11.1)	42.0 (11.8)
	*p* = 0.84	*p* = 0.80	***p* = 0.04**	*p* = 0.77
**Race ****				
White	51.2 (9.2)	53.7 (10.7)	54.4 (10.4)	42.3 (10.7)
Non-White	50.6 (9.6)	52.5 (10.8)	56.6 (10.4)	41.0 (11.1)
	*p* = 0.75	*p* = 0.24	***p* < 0.01**	***p* < 0.01**
**Ethnicity ****				
Hispanic or Latino	53.5 (10.9)	52.4 (7.7)	55.1 (8.2)	43.8 (9.4)
Not Hispanic/Latino	50.9 (9.3)	53.3 (10.8)	55.3 (10.5)	41.7 (10.9)
	*p* = 0.26	*p* = 0.55	*p* = 0.73	*p* = 0.45
**Marital status ****				
Married	50.4 (9.0)	52.6 (10.7)	54.1 (10.4) ^s^	42.7 (11.4) ^so^
Single	51.4 (9.8)	54.5 (10.8)	56.8 (10.4) ^m^	40.8 (10.0) ^m^
Other	51.3 (9.3)	54.2 (10.5)	56.9 (9.7)	38.1 (10.3) ^m^
	*p* = 0.19	*p* = 0.06	***p* < 0.01**	***p* < 0.01**
**Insurance status ****				
Medicare	51.5 (9.5)	54.2 (10.1)	55.5 (10.6) ^p^	39.5 (10.4) ^p^
Private	50.2 (9.0)	52.2 (11.3)	53.8 (10.1) ^md^	44.7 (10.9) ^md^
Medicaid	52.4 (9.5)	54.5 (10.9)	58.0 (9.7) ^p^	40.3 (11.2) ^p^
Other	51.1 (8.9)	53.1 (9.5)	57.3 (6.0)	38.4 (11.3)
	*p* = 0.06	*p* = 0.10	***p* < 0.01**	***p* < 0.01**
**Online portal use ****				
Activated	50.4 (9.0)	52.5 (10.3)	54.1 (10.1)	43.1 (10.4)
Other	52.9 (10.2)	57.4 (11.7)	59.7 (10.8)	36.5 (11.5)
	***p* < 0.01**	***p* < 0.01**	***p* < 0.01**	***p* < 0.01**
**CCI score ***	0.03 (−0.03, 0.10)	0.16 (0.09, 0.23)	0.08 (0.01, 0.15)	−0.21 (−0.28, −0.14)
	*p* = 0.30	***p* < 0.01**	***p* = 0.03**	***p* < 0.01**
**Median income ***	−0.04 (−0.10, 0.02)	−0.04 (−0.11, 0.03)	−0.13 (−0.20, −0.07)	0.11 (0.04, 0.18)
	*p* = 0.19	*p* = 0.25	***p* < 0.01**	***p* < 0.01**
**Unemployment rate ***	0.04 (−0.03, 0.10)	0.02 (−0.05, 0.09)	0.12 (0.05, 0.19)	−0.10 (−0.17, −0.03)
	*p* = 0.26	*p* = 0.52	***p* < 0.01**	***p* < 0.01**

* Continuous variables are reported as Spearman’s correlation coefficient and 95% confidence interval in parentheses. ** Categorical variables are reported by using mean PRO score and standard deviation in parentheses. Superscripts reflect significant mean difference in post-hoc *t*-tests of marital status (^m^, married; ^s^, single; ^o^, other) and insurance (^p^, private; ^m^, Medicare; ^d^, Medicaid). Bold represents *p*-values less ≤ 0.05.

**Table 3 cancers-16-01015-t003:** Univariable analyses of tumor and treatment characteristics vs. PRO scores. Henry Ford Health, Detroit, Michigan, January 2020 to December 2022.

Variable	Depression	Fatigue	Pain Interference	Physical Function
**Primary disease site ***				
Breast	49.7 (8.6) ^ghlp^	51.1 (10.0) ^glo^	53.0 (9.3) ^ghlp^	44.4 (10.0) ^glo^
Head and neck	52.6 (8.9) ^bop^	53.1 (10.5) ^gp^	58.4 (10.4) ^bop^	42.3 (11.4) ^gp^
Lung	52.8 (9.5) ^bop^	55.6 (10.9) ^bp^	57.7 (10.8) ^bop^	37.8 (10.6) ^bhp^
Gastrointestinal	51.7 (9.3) ^bp^	56.2 (10.3) ^bhp^	56.5 (10.4) ^bop^	39.5 (10.3) ^bhp^
Prostate	45.8 (9.9) ^bghlo^	46.7 (12.4) ^ghlo^	45.6 (7.7) ^bghlo^	47.5 (10.5) ^ghlo^
Other	50.2 (9.9) ^hlp^	55.0 (10.9) ^bp^	53.9 (10.8) ^ghlp^	40.4 (11.5) ^bp^
	***p* < 0.01**	***p* < 0.01**	***p* < 0.01**	***p* < 0.01**
**Stage ***				
Stage 0	49.0 (10.9) ^4^	48.9 (9.4) ^234^	52.7 (10.2) ^4^	44.2 (11.5) ^4^
Stage 1	50.5 (8.9) ^4^	51.2 (9.8) ^4^	54.0 (9.7) ^4^	44.5 (10.1) ^4^
Stage 2	49.9 (8.3) ^4^	53.1 (10.8) ^04^	54.4 (10.1) ^4^	42.3 (10.7) ^4^
Stage 3	50.2 (9.5) ^4^	53.5 (10.3) ^04^	54.6 (11.2) ^4^	42.0 (11.2) ^4^
Stage 4	53.5 (9.2) ^0123^	57.0 (11.6) ^0123^	58.7 (10.1) ^0123^	37.8 (10.7) ^0123^
	***p* < 0.01**	***p* < 0.01**	***p* < 0.01**	***p* < 0.01**

* Categorical variables are reported by using mean PRO score and standard deviation in parentheses. Superscripts reflect significant mean difference in post hoc *t*-tests of primary site (^b^, breast; ^g^, gastrointestinal; ^l^, lung; ^h^, head and neck; ^o^, other; and ^p^, prostate) and stage (^0^, ^1^, ^2^, ^3^ and ^4^). Bold represents *p*-values less ≤ 0.05.

**Table 4 cancers-16-01015-t004:** Multivariate analyses related to PRO scores. Henry Ford Health, Detroit, Michigan, January 2020 to December 2022.

Depression Score		
Variables	Mean Score Change (95% CI)	*p*-Value *
Unemployment rate	1.03 (−6.29, 8.35)	0.78
Stage of disease **		
Early disease (Stages 0–2)	-	-
Advanced disease (Stages 3–4)	1.96 (0.65, 3.27)	**<0.01**
**Fatigue score**		
**Variables**		
Unemployment rate	−6.04 (−16.96, 4.89)	0.28
Stage of disease **		
Early disease (Stages 0–2)	-	-
Advanced disease (Stages 3–4)	3.94 (2.16, 5.72)	**<0.01**
Insurance status ^+^		
Private	-	-
Medicaid	1.71 (−1.16, 4.57)	0.24
Medicare	1.48 (−0.24, 3.21)	0.09
Other	−0.45 (−7.17, 6.26)	0.90
**Pain Interference score**		
**Variables**		
Unemployment rate	9.11 (−0.04, 18.26)	0.051
Stage of disease ^+^		
Early disease (Stages 0–2)	-	-
Advanced disease (Stages 3–4)	1.33 (−0.52, 3.19)	0.16
Primary disease site **		
Breast		
Head and neck	4.36 (2.29, 6.43)	**<0.01**
Lung	2.73 (0.17, 5.29)	**0.04**
Gastrointestinal	1.84 (−0.67, 4.35)	0.15
Prostate	−8.73 (−13.47, −3.99)	**<0.01**
Other	−0.53 (−2.76, 1.69)	0.64
Online portal use	−4.50 (−6.38, −2.61)	**<0.01**
**Physical function score**		
**Variables**		
Unemployment rate	−5.40 (−16.44, 5.64)	0.34
Stage of disease **		
Early disease (Stages 0–2)	-	-
Advanced disease (Stages 3–4)	−3.13 (−4.99, −1.26)	**<0.01**
Insurance status **		
Private	-	-
Medicaid	−3.14 (−6.15, −0.13)	**0.04**
Medicare	−3.20 (−5.64, −0.75)	**0.01**
Other	−3.06 (−9.98, 3.85)	0.39
Age	−0.08 (−0.17, 0.02)	0.11
CCI score	−0.68 (−1.07, −0.29)	**0.01**

* *p*-value for individual estimate (Z statistic); ** likelihood ratio test, *p* ≤ 0.01; ^+^ likelihood ratio test, *p* > 0.05. Bold represents *p*-values less ≤ 0.05.

## Data Availability

The data presented in this study are available upon request from the corresponding author.

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
