# Peer review of "Association of Social Determinants with Patient-Reported Outcomes in Patients with Cancer"

_cancers, 2024, doi:10.3390/cancers16051015_

Round 1
Reviewer 1 Report
Comments and Suggestions for Authors
The manuscript entitled „ Association of Social Determinants with Patient Reported Outcomes in Patients with Cancer „ deals with a very important topic. Getting patient-reported outcome (PRO) measures gives substantial information to the treating personnel beyond what the doctors or nurses can perceive from personal communication. It is also important to know whether there is a connection between PRO values and other circumstances. In this paper, the authors investigated what is the association between PROs and social determinants. They have found that several social factors have an impact on PRO values. It is quite interesting and important for treatment adjudication where the patient's quality of life is a significant factor. That is why I think this work is valuable, although it was performed in only one institution, there were missing data and the sample size of some subgroups was rather modest.
I have only a few remarks or comments:
1. In the Materials and Methods section (row 90) they wrote “patient was included if the PRO survey was completed within 6 months of the initial date of diagnosis.” In my opinion, it needs an explanation (rationale) why this time threshold had been chosen and a supplement on whether the treatments or interventions started during these 6 months affect PRO values.
2. The distribution of primary disease sites is uneven which may affect the outcome of this investigation. It could not be ruled out that the effect of social determinants is different according to different tumors. Some sample size is really small (prostate cancer 2.2 %). I think it also deserves a comment in the discussion section.
Reviewer 2 Report
Comments and Suggestions for Authors
The manuscript focuses on the analysis of associations between patient reported outcome scores and social determinants of health among patients newly diagnosed with cancer, important subject in the context of providing healthcare for cancer patients.
However, there are some parts of the manuscript that could be improved, e.g., the introduction, the description of methods, and the discussion. The simple summary and abstract repeat major part of the information on the study with slight differences. In addition, the description of the study provided in both is very odd, including very short sentences that result in a fragmented synthesis of the paper.
The Introduction presents very briefly the use of patient-reported outcome scores for cancer patients, and the role of social determinants of health in influencing cancer patients' perceptions. Thus, it would be interesting to improve the introduction by incorporating more robust literature review on both subjects.
The Methods section lacks information on characteristics and location of the institution where patients were enrolled, in addition to details on the application of the survey, type of questionnaire, information collected, data processing, and categorization of variables of the study.
The graphs presented in the Results (Figures 1-3) do not contribute to the description of the study, considering that variables on educational attainment were neither included in the tables nor in the univariate or multivariate analysis. In addition, tables titles and captions lack proper information to synthesize their respective contents.
Finally, the Discussion should address the issue of performing the survey during the period of COVID-19 pandemics, which could have influenced the results of the study.
Comments on the Quality of English LanguageThe quality of English language is fine.
Round 2
Reviewer 2 Report
Comments and Suggestions for Authors
The second version of the manuscript included changes in the Introduction, Methods and Discussion; however, there is still lack of information on the location of the study (i.e., at least the country of location should be indicated to allow that the study could be analyzed and included in systematic literature reviews and other types of citations).
In addition, the authors failed to exclude the figures 1-3, considering that their presentation does not bring relevant information to the manuscript. The information on educational attainment should be presented in the tables, according to my previous comments, instead of including graphs that do not allow identification of the variable distribution (Table 1) or its associations with PRO scores (Table 2).
There is also lack of comments in the Results and in the Discussion on the absence of associations between PRO scores and educational attainment of patients. Thus, maybe the variable referring to educational attainment and the graphs could be totally excluded from the paper, since the information is not used anywhere in the analysis.
Finally, authors included very odd sentences in the titles of tables (e.g., "Table 1. Patient demographics. There was a diverse patient population, with non-White patients representing 29.6% of the cohort. The majority of patients did not have private insurance, and nearly 80% of patients utilized the online portal to complete the survey").
The reviewers may have asked to either include additional comments on the results obtained or modify titles of the tables to be more informative; however, it seems that the authors confused both requests.
In sum, tables should have informative titles that inform readers on the tables contents independently of the text (e.g., "Table 1. Sociodemographic and health characteristics of patients in the study. [Location of data collection], [Year of data collection]."), while the text of the section Results before the table should include additional comments on the data (e.g., "A total of 7,285 recorded surveys were completed by 4,016 unique patients. Of this group, 1,090 unique patients completed a survey within the first 6 months after their initial diagnosis and were included in the study (Table 1). Mean age was 60.2 ± 12.75 years. Female patients composed 60.3 percent of the cohort. Non-White patients represented 29.6 percent of the cohort. There was a diverse patient population, with non-White patients representing 29.6% of the cohort. The majority of patients did not have private insurance, and nearly 80% of patients utilized the online portal to complete the survey.").
The same should be applied to Table 2; i.e., title of the table should be: "Table 2. Univariable analyses of patients' sociodemographic characteristics in relation to PRO scores. [Location of data collection], [Year of data collection]."; while the text before the table should be "Interestingly, patients who used the online portal to complete their survey had significantly improved individual PRO scores in each domain compared to those who completed their survey at their clinic visit (t-test, p < 0.01). Married patients and patients who utilized the online portal were among the patients to have significantly improved PRO scores.".
Comments on the Quality of English LanguageThe quality of English language is fine.
Round 3
Reviewer 2 Report
Comments and Suggestions for Authors
The revised version of the manuscript is fine, included suggestions mentioned in previous reviewers' comments.